# Detection of Ancient Viruses and Long-Term Viral Evolution

**DOI:** 10.3390/v14061336

**Published:** 2022-06-18

**Authors:** Luca Nishimura, Naoko Fujito, Ryota Sugimoto, Ituro Inoue

**Affiliations:** 1Human Genetics Laboratory, National Institute of Genetics, Mishima 411-8540, Japan; rnishimura@nig.ac.jp (L.N.); nfujito@nig.ac.jp (N.F.); rsugimot@nig.ac.jp (R.S.); 2Department of Genetics, School of Life Science, The Graduate University for Advanced Studies (SOKENDAI), Mishima 411-8540, Japan

**Keywords:** ancient DNA, ancient RNA, PCR, NGS, ancient virus, ancient virome, bioinformatics, viral evolution, TDRP

## Abstract

The COVID-19 outbreak has reminded us of the importance of viral evolutionary studies as regards comprehending complex viral evolution and preventing future pandemics. A unique approach to understanding viral evolution is the use of ancient viral genomes. Ancient viruses are detectable in various archaeological remains, including ancient people’s skeletons and mummified tissues. Those specimens have preserved ancient viral DNA and RNA, which have been vigorously analyzed in the last few decades thanks to the development of sequencing technologies. Reconstructed ancient pathogenic viral genomes have been utilized to estimate the past pandemics of pathogenic viruses within the ancient human population and long-term evolutionary events. Recent studies revealed the existence of non-pathogenic viral genomes in ancient people’s bodies. These ancient non-pathogenic viruses might be informative for inferring their relationships with ancient people’s diets and lifestyles. Here, we reviewed the past and ongoing studies on ancient pathogenic and non-pathogenic viruses and the usage of ancient viral genomes to understand their long-term viral evolution.

## 1. Introduction

Ancient DNA and RNA are nucleic acids derived from archaeological or historical remains, including bones, teeth, mummified tissues, and coprolites. The first ancient DNA study was published in 1984. It demonstrated the 229 bp mitochondrial DNA fragments from the dried muscle of an *Equus quagga* specimen preserved in a museum by cloning into the λ gt10 vector [1]. Many similar studies have been reported using recent technological developments, including the polymerase chain reaction (PCR) and next-generation sequencing (NGS) techniques. Specifically, NGS has allowed us to obtain a massive quantity of ancient genomic data, mostly targeting the genomes of archaic humans. Therefore, we can estimate past events, including gene flow between different human species, migration history, and population structures [2,3,4,5,6,7]. According to ancient human genomic studies, several research groups have discovered that bacteria and viruses co-existed in ancient people’s bodies as they do in modern ones [8,9,10,11]. Most studies have focused on the pathogenic bacteria or viruses that previously caused pandemics or epidemics. For instance, the first ancient viral study was published in 1997 on the influenza virus that caused the Spanish influenza pandemic in 1918 [12]. Subsequently, several pathogenic bacterial and viral genomic sequences have been discovered from ancient samples, including *Yersinia pestis* and Hepatitis B virus [13,14,15]. The pathogenic data from ancient times are informative in inferring past pandemics and epidemics [16,17]. Essentially, we can directly examine the differences between modern and ancient viral genomes using ancient viral sequences. It would be helpful to understand viral evolutionary history across thousands of years. Although evolutionary estimation with only modern samples can detect short-term evolutionary processes with population-level processes, ancient samples can provide information on long-term viral evolution, reflecting the actual fixation ratio [18]. The expansion of the number of ancient viral genomes allows for the elucidation of the viral evolution of various viral species. Understanding long-term viral evolution might be beneficial for reconstructing epidemic history and possibly predicting future trends. Furthermore, the analyses of viral components or viromes in ancient people’s bodies might be informative for inferring the lifestyle of the ancient individuals. A virome is defined as all of the viral assemblages existing in a given environment, including the human gut, oral environment, and skin [19]. The human virome consists of the viruses infecting the eukaryotic cells, archaea, and bacteria in the human body as well as transient viruses in food [20,21]. It includes the commensal, opportunistic, and pathogenic viruses which cause a broad range of immune responses, some of which cause chronic infections [22]. 

Here, we review the ancient viral studies that made use of ancient materials over decades based on the following topics: history, samples, data analyses, ancient virome, specific ancient viral examples, and long-term viral evolution.

## 2. History of Ancient Viral Studies

Paleovirology is the study of ancient viruses derived from ancient samples including extinct viruses [23,24]. Ancient DNA and RNA sequence data have been used in studies of ancient viruses from historical samples approximately two decades old to tens of thousands of years old. The first ancient viral study, which analyzed the 1918 Spanish influenza virus, was reported in 1997 [12]. Since then, various ancient viruses have been discovered, as shown in Figure 1. In the early stages in the 2000s, ancient viral studies were conducted using the PCR technique [12,25,26]. The PCR technique is convenient but cannot be applied to unknown sequences or highly diverged sequences. It amplifies only specific targeted regions, which can lead to the invalid estimation of viral evolution, such as with respect to root ages. The NGS technique appeared in 2005, allowing us to obtain a vast quantity of genomic data [27]. An ancient DNA study using the NGS technique was published in 2006 showing a wooly mammoth’s genomic sequences [28]. Since then, it has been applied to ancient viral studies and resulted in the accumulation of viral genomic information. NGS data, such as whole genomic sequencing (WGS) data and the capture-based sequencing of particular cases, can provide us with a global picture of ancient viral genomic sequences, which means that viral evolution can be estimated more precisely. Thus far, ancient viral studies have mostly focused on specific pathogenic viruses because of their public health importance and the availability of reference genomes. Additionally, there have been several studies on non-pathogenic ancient viruses including non-human viruses [29,30]. Thanks to developments in metagenomics, viral genomic data are accumulating using materials in a specific environment with NGS techniques, leading to virome research [9].

## 3. Samples for Ancient Virus Discoveries and How to Analyze Them

### 3.1. Historical Samples

Ancient viruses exist in the remains of ancient organisms in which viral hosts co-existed. Mineralized tissues including bones, coprolites, and teeth are useful in the recovering of viral sequences preserved for thousands of years. Mineralized tissues are more abundant and highly preserved, allowing large-scale population sampling [31]. They are suitable for specific viral sequence analyses and commensal viruses. For example, 14th-century coprolites were used for virome analyses characterizing the viral community of the gut environment [9]. Conversely, soft tissues cannot be preserved longer than mineralized tissues and have serious limitations when it comes to their sample ages [32]. There are advantages when using soft tissues, such as formalin-embedded soft tissue, which can be preserved in museums or institutes for a period ranging from a number of decades to 200 years, and mummified tissue, where pathological lesions can be visible and from which pathological viruses can be extracted [12,33]. Overall, both historical soft and mineralized tissue provide various kinds of ancient DNA and RNA viruses. 

For plant viral studies, barley grains, corn cobs, potato, and herbarium have been used, and RNA and DNA preservation was displayed [29,34,35,36,37,38,39,40,41,42]. Non-organismal samples such as ice cores have been effectively used to reconstruct ancient viromes and obtain replication-competent DNA viruses [43,44,45,46]. Cultural artifacts such as birch pitches may also contain viral sequences related to humans. For example, a 5700-year-old birch pitch reflecting an ancient human oral environment indicated the existence of the Epstein–Barr virus within the microbiome [47]. Vaccinia virus, one of the orthopoxviruses, has also been discovered on vaccination kits from the 1860s, indicating vaccination materials at the early stage of vaccination development [48,49,50].

Due to hydrolytic depurination and subsequent β elimination, ancient DNA has typically experienced single-strand breaks and fragmentation [51,52]. Moreover, the other prominent event is cytosine deamination, which involves cytosine turning into uracil or thymine when cytosine is methylated. Since it occurs at the overhanging ends of ancient DNA fragments, the C to T substitution ratio increases towards the ends of sequencing reads when mapped against reference sequences [51,52,53]. The postmortem degradation rate depends on the temperature and humidity. One report of an in situ DNA decay assay with different temperatures demonstrated that DNA fragmentation can occur rapidly at high temperatures [54]. Regarding RNA, cytosine can also be deaminated like DNA [29]. Nevertheless, different damage susceptibilities are invulnerable to depurination and depyrimidination and vulnerable to hydrolysis [55]. Weaker phosphodiester links and stronger N-glycosidic bonds cause these susceptibilities [55]. Either way, samples preserved in excellent conditions such as at a low temperature and low humidity together with the preservation of external morphology are preferred in the context of obtaining high-quality sequencing data.

### 3.2. DNA and RNA Extraction, Amplification, and Sequencing

Contaminated exogenous nucleic acids and co-extracted small molecules should be avoided to obtain sufficient DNA or RNA. Since ancient endogenous DNA is highly fragmented and exhibits low abundance, contaminations dramatically affect the DNA yields and inhibit the downstream enzymatic reactions [52,56]. To avoid these contaminations, several ancient DNA extraction methods have been developed for various kinds of historical samples, such as bone and teeth [56,57,58]. Modern contamination can also be distinguished based on the presence or absence of postmortem damage patterns with bioinformatic analyses as described in the next section. Several methods introduced USER reagent (New England Biolabs, Ipswich, MA, USA), which is a commercialized enzymatic mix of uracil-DNA-glycosylase (UDG) and endonuclease VIII (Endo VIII) and can remove uracil residues and repair the resulting abasic sites [52,59]. USER has been utilized to eliminate damage patterns and avoid damage-induced sequence errors. It has been used for ancient DNA authentication. In the case of mammalian nuclear DNA, the damage pattern can be still detected by examining CpG dinucleotides [52]. When we use non-mammalian DNA, we can build both USER-treated (full-UDG) and non-treated (non-UDG) libraries for sequencing or use a modified method (UDG-half) that restricts the ancient DNA damage at the terminal nucleotide [52,60]. The usage of USER treatment depends on the study, including the study design and research question.

In early ancient viral studies, PCR techniques were applied to amplify ancient DNA sequences derived from the target viral species using the specific primer sets followed by Sanger sequencing. However, those techniques detect specific or limited genomic regions and cannot be applied to detect unknown viral sequences or metagenomic analysis. For example, recombination and different gene contents might hinder fragment amplification by PCR. If the DNA experience massive postmortem damage, such as fragmentation and substitution, it might be challenging to detect ancient viral genomes using PCR with specific primers.

WGS and capture-based sequencing using NGS techniques are more informative solutions to overcoming the above problems. NGS platforms are helpful when studying viral genomes due to their capacity to obtain sequence data across the whole genome and a representative fraction of the overall population [61,62,63]. Metagenomic analyses with untargeted WGS data are a powerful means of detecting ancient viruses more comprehensively. For example, several ancient viral sequences, including the hepatitis B virus (HBV) and Siphovirus contig89, have been discovered from WGS data generated for genomic analyses of ancient people [14,30]. Moreover, WGS data has the potential to provide various kinds of viral genomic information because of a vast quantity of sequence data. However, a low copy number of DNA or poor preservation quality might cause difficulties in reconstructing complete ancient viral genomes despite the use of WGS [64,65]. It has been shown that capture-based sequencing enrichment of the target viral sequences, followed by NGS data analyses, possibly recovers the complete genomes of ancient viruses [14,66,67].

### 3.3. Bioinformatic Analysis

Multiple bioinformatic tools are publicly available for the analysis of sequencing data. These tools provide basic protocols for quality control, searching, and the genome assembly of the subject sequencing data. Figure 2 shows the overall procedure of the analyses for ancient viral sequences. For WGS data, the preprocessing of the output reads data is essential. For instance, adapters, contaminations, low quality reads, and duplications need to be removed using trimming tools, including AdapterRemoval, Trimmomatic, Picard Tools, and BBTools [68,69,70,71]. leeHom is designed for ancient DNA data and can be utilized in adapter trimming and merge reads based on the Bayesian maximum a posteriori probability approach [72]. Reads aligned to the human genome can also be removed to lower the computational burden, as shown in several ancient viral studies [30,73]. However, these human decontamination steps might remove several viral sequences with high homology with a part of the human reference genome such as the herpesvirus sequence. Therefore, an appropriate analytical design should be chosen based on the researcher’s purpose, and the caveat of the decision can be described in the method. The remaining reads could be used for the downstream analysis, as shown in Figure 2. If the ancient DNA has been highly preserved and experienced only a small number of fragmentation events, then de novo assembly could be an option for obtaining the longer sequences or contigs using assembling tools, such as SPAdes and MEGAHIT [74,75,76]. Assembled contigs might be helpful in detecting highly diverged ancient viral genomes which might not exist in the present reference genomes.

Many ancient samples have suffered from the contamination of modern sequences, resulting in metagenomic datasets mixed with ancient and modern sequences [77]. Borry et al. created a robust and automated approach for ancient DNA damage estimation and authentication of de novo assembled ancient DNA, PyDamage [77]. Because of the high degradation and fragmentation of ancient DNA, efficient de novo assembly is challenging. In such a case, sequence binning, a sequence clustering procedure that distributes the sequences into individual groups, can be performed as shown in Figure 2. The binning can be either taxonomy-dependent binning (taxonomy binning) or taxonomy-independent binning (genome binning) [78]. Taxonomy binning is a supervised method based on known genomic sequences such as DIAMOND and MetaPhlAn2 (see the primer on tools for taxonomic classification written by Ye et al.) [79,80,81,82]. Genome binning is an unsupervised method which uses machine-learning methods according to the feature patterns of sequences and linkage patterns such as CONCOCT and MetaBAT2 (see the review about tools for taxonomy-independent binning written by Sedlar et al.) [83,84,85]. Binning tools for viral genomes have been developed, such as phages from metagenomics binning (PHAMB) [86]. Arizmendi Cárdenas et al. reported that Centrifuge would be the most suitable tool for the identification of human DNA virus from ancient samples based on a simulation, with it demonstrating the highest sensitivity and precision when compared with other taxonomic binning tools such as Kraken2, DIAMOND, and MetaPhlAn2 [87,88,89]. Such tools have already been applied in several ancient microbiome studies and might be also useful for ancient viral genome classification and reconstruction [11,90]. In addition to sequence binning, preprocessed reads, PCR amplified fragments, and assembled contigs can be aligned to viral reference sequences using several alignment tools that detect ancient viral sequences, including the Basic Local Alignment Search Tool (BLAST), Burrows–Wheeler aligner (BWA), and MEGAN Alignment Tool (MALT) [91,92,93,94]. Viral genomes on databases such as RefSeq and IMG/VR can be used as reference viral genomes [95,96]. Pratas et al. established the first NGS pipeline, TRACESPipe, for identifying, analyzing, and assembling viral DNA by combining data from multiple organs [97]. TRACESPipe utilizes hybrid methodological approaches with reference-based and de novo assembly and has the potential to reconstruct and analyze ancient viral genomes from various kinds of ancient samples. Pipelines for ancient DNA analyses such as PALEOMIX, FALCON-Meta, and nf-core/EAGER might also prove helpful in the discovery of ancient viral genomes [98,99,100]. Such pipelines integrate several steps such as read preprocessing, genome alignment, ancient DNA authenticity, genotyping, and metagenomic profiling. PALEOMIX was utilized for an ancient oral microbiome study carried out by Jensen et al., and they detected fragments of the Epstein–Barr virus [47]. FALCON-Meta was applied for metagenomic analyses using a 110,000- to 130,000-year-old tooth of a polar bear from Svalbard, but Patas et al. themselves claimed that the observed viral genomes could be contaminated sequences such as Parvovirus [99,101].

The obtained ancient viral genomes can be utilized for further analyses such as genome characterization and phylogenetic analyses. Ancient viral genomes can be characterized by several tools for gene annotation and recombination analysis such as BLAST, Prodigal, and RDP4 [102,103]. For phylogenetic analyses, we use multisequence alignment data generated by software such as MAFFT. The data can be applied for the construction of neighbor joining, maximum likelihood (ML), and Bayesian phylogenetic trees by using the following tools: MEGA, PhyML, RAxML, MrBayes, BEAST, and BEAST2 [104,105,106,107,108,109,110].

It should be confirmed whether the detected ancient viral sequences exhibited postmortem damage, including fragmentation and cytosine deamination. Since modern contamination from a soiled laboratory environment is a serious threat to ancient viral studies, and the experimental environment could be a severe threat to ancient viral studies, ancient sequences can be distinguished from modern ones using damage patterns. Several analytical methods such as mapDamage, DamageProfiler, and PMDtools have been established for detecting damage [111,112,113,114]. These tools plot ancient DNA damage patterns and visualize the frequency of particular base misincorporations. The visualization of C to T substitution frequencies at the 5′ end is helpful in ancient DNA authentication. PMDtools can calculate the postmortem damage (PMD) score to distinguish the genuinely ancient sequence from modern contamination using a likelihood model incorporating PMD, base quality scores, and biological polymorphism [113].

## 4. Metagenomic Data to Comprehend the Ancient Human Virome

Viromes have been analyzed from the vast quantity of shotgun sequencing data to identify (ideally) complete but (most commonly) partial viral genomes. Several studies revealed a highly diverged viral population across the human body and individuals in modern samples as well as those associated with several factors, including diet, age, geographic location, and disease status [115,116,117,118,119,120,121,122,123]. The distinct characteristics of viromes can be observed among modern and ancient human bodies. Metagenomic sequencing data derived from ancient samples can help to detect such differences. The first ancient virome study was published in 2014 and it analyzed 14th-century coprolites and paleofeces in Belgium [9]. The viral particles were first isolated from the samples through filtration, and high-throughput pyrosequencing was conducted using the 454 Life Science Genome FLX sequencer, one of the earliest NGS platforms. Eukaryotic, archaeal, and bacterial viruses were detected from the metagenomic data, and *Siphoviridae* phages dominated within the detected viral reads, with approximately 60% abundance within known viral reads. Differences existed between the coprolite virome and modern human stool virome at the taxonomic level, but they were functionally more similar based on principal component analysis using taxonomical or functional annotation [9]. This suggested that the functional roles of viromes might be conserved between ancient and modern gut environments, which was consistent with a previous report demonstrating the significant conservation of gene contents within viromes of the same ecological niche despite individual taxonomic variability [124]. Viral communities might play a significant role in maintaining gut environments.

Another virome study in which ancient DNA was extracted from three pre-Columbian Andean mummies was published in 2015, and the viral components were analyzed [125]. Metagenomic sequencing was conducted using the Illumina MiSeq platform, and viral sequences were identified through homology analyses. Bacteriophages dominated within the reads as follows: 50.4% (mummy FI3), 1.0% (mummy FI9), and 84.4% (mummy FI12). Those phages derived from the *Siphoviridae*, *Myoviridae*, *Podoviridae*, and *Microviridae* families, respectively, and their probable host bacteria were predicted. The results suggested that the natural mummification of the human gut preserved viral DNA, which was utilized to infer the ancient gut virome.

Wibowo et al. analyzed eight 1000- to 2000-year-old human coprolites and reconstructed high-quality ancient microbial genomes [126]. The recovered microbial components were compared with present-day human gut microbiomes, demonstrating that the palaeofaeces microbiomes were more similar to non-industrialized human gut microbiomes than industrialized ones. Differences in dietary habits might be a factor in determining such differences. Rampelli et al. illustrated that modern virome data obtained from urban societies, hunter-gatherer communities, and pre-agricultural tribes demonstrate different viral contents in each cultural group [127].

So far, we have obtained only limited amounts of ancient virome information because this field is still in its infancy. The accumulation of ancient virome data might elucidate viral evolution in a more precise way.

## 5. Specific Ancient Viral Studies Inferring Past Pandemic and Evolutionary History

The benefits of studying ancient viruses include detecting the predation of the human pathogens that caused pandemics in ancient times and learning how viruses have evolved over thousands of years. Several studies have discovered ancient viral sequences from historical samples, as shown in Table 1. Most of them are human pathogenic viruses because they are related to past pandemics and are epidemiologically important. However, focusing on only human pathogenic viruses in ancient viral studies could be biased in terms of the evolution of viruses because their evolution has been affected by immune responses in the human body and adaptation to humans. Non-pathogenic and non-human viruses might reflect co-evolution with different host species and show diverged viral evolution. Therefore, ancient viral studies have been conducted on non-human and non-pathogenic viruses.

Various epidemics and pandemics have been recorded or assumed to have occurred throughout human history [16]. Ancient viral genomes have given insights into the pandemics and epidemics of the past and virus–human interactions on an evolutionary timescale. For example, the hepatitis C virus (HCV), papillomaviruses, anellovirus, tomato mosaic tobamovirus (ToMV), and T-lymphotropic virus have been detected through PCR amplification of targeted regions, as shown in Table 1 [128,134,137,139,150,151,152].

Several studies successfully obtained complete ancient viral sequences using the PCR technique such as influenza viruses in 1918 and human immunodeficiency virus type 1 (HIV-1) in the 1970s. First, the 1918 influenza viral RNA was amplified using reverse transcription polymerase chain reaction (RT-PCR) from formalin-fixed paraffin-embedded or frozen lung tissue specimens and sequenced. The complete coding sequences were determined by several studies. The 1918 virus might have originated from an avian source and adapted to mammals with several mutations [12,142,143,144,145,146,147,148]. The complete coding sequence of the 1918 influenza virus was cloned and applied for infectious experiments to characterize its virulence, with it displaying a high-proliferation phenotype in human bronchial epithelial cells [161]. In the case of HIV-1, an almost complete genome was obtained from the serum of 1970s samples in the US and formalin-fixed paraffin-embedded tissues from Kinshasa, Democratic Republic of Congo, in 1966 [135,136]. Worobey et al. illustrated that the US HIV-1 epidemic showed extensive genomic diversity and emerged from the pre-existing Caribbean epidemic [135]. They also suggested that HIV-1 entered the US around 1970, and New York City was the critical center of early US HIV diversification based on their phylogenetic analyses. Gryseels et al. reconstructed the oldest HIV-1 genome. The new phylogenetic tree, including the oldest genome, revealed that the origin of the pandemic lineage of HIV-1 dated to approximately the turn of the 20th century [136].

A 2.2 kb complete ancient caribou feces-associated virus (aCFV) was reconstructed using PCR and characterized from 700-year-old caribou feces frozen in a permanent ice patch [43]. The aCFV was estimated to be distantly related to the plant-infecting geminiviruses and the fungi-infecting Sclerotinia sclerotiorum hypovirulence-associated DNA virus 1 and derived from plant material ingested by caribou. The infectivity of the reconstructed viral genome was confirmed with the model plant *Nicotiana benthamiana*. Older permafrost layers also revealed the existence of replication-competent viruses such as 30,000-year-old *Pithovirus sibericum* and *Mollivirus sibericum*, also known as a giant virus [44,45]. Both of the viruses in the caribou feces and the 30,000-year-old virus demonstrated their infectivity, and they did not show postmortem degradation patterns. Frozen encapsidated viruses might have escaped from ancient DNA damage [129].

In contrast to the above successes, short and limited PCR products might cause inaccurate estimation results in relation to viral evolution. For example, the most recent common ancestor (MRCA) of human parvovirus B19 (B19V) was estimated to be from the early 1800s based on 121 and 154 bp PCR amplified fragments from the 70-year-old bones of Finnish soldiers [138]. However, 500- to 6900-year-old B19V viruses were detected in a subsequent study, and the estimated MRCA was inconsistent [73]. The new result suggested that the MRCA was placed ~12,600 years ago, with a substitution rate of approximately 1.22 × 10^−5^ substitutions/site/year, which is lower than the previous estimate of 2.1–2.2 × 10^−4^ substitutions/site/year calculated with 70-year-old B19V fragments [73,138]. Another study using ancient B19V sequences detected from dental samples between the 16th and 18th centuries supported this estimation [67]. Ancient human T-cell leukemia virus type 1 (HTLV-1) fragments with a length of about 160 bp were detected from a 1500-year-old Andean mummy, leading to the conclusion that HTLV-1 was carried by ancient Mongoloids to the Andes before the Colonial era [139]. However, it has been suspected that the sequence was derived from modern contamination based on the current prevalence of HTLV-1 in American Indians and molecular clock calculation [140,141]. Since DNA and RNA fragmentation likely results in fragments of fewer than 100 bp, it is difficult to amplify longer sequences.

Since the appearance of NGS techniques, several ancient viral genomes have been detected and analyzed by NGS techniques, as described in Table 1. The oldest ancient RNA virus discovered from WGS data is the measles morbillivirus (MeV), which was extracted from a formalin-fixed lung specimen collected in 1912 in Berlin [149]. The quasi-complete MeV genome was constructed with a mean coverage of 54x derived from Illumina platforms. The Bayesian and maximum likelihood phylogenetic trees suggested that the MeV and rinderpest viruses potentially diverged around 2500 years ago and was followed by cattle-to-human host transmission, with the subsequent evolution of two distinct lineages occurred [149]. The estimation seemed to be consistent with demographic changes in the past. The viral divergence event coincided with an increase in the population size, which became larger than the MeV critical community size (CCS) of 250,000, to 500,000 individuals, supporting the continuous transmission of MeV [162,163,164].

Concerning the reverse transcribing of viruses, the hepatitis B virus (HBV) has been well-studied using WGS data. The HBV genome is relatively small, approximately 3 kb; thus, its genome can be easily reconstructed. The first ancient HBV study in 2012 confirmed the existence of ancient HBV in a Korean mummy around the 16th century [33]. The constructed genotype C2 HBV sequence, commonly spread in Southeast Asia, suggested that the Korean HBV sequence dated back at least 3000 to 100,000 years based on its genetic diversity when compared with modern samples. Subsequently, several groups have discovered ancient HBV genomes from ancient samples: a 500-year-old Italian mummy, a 2000-year-old Egyptian mummy, 500-year-old Mexican teeth, and 400- to 105,000-year-old Eurasian and American teeth and bones, as listed in Table 1 [14,15,67,130,131,132,133]. These samples were analyzed with WGS or the capture-based sequencing data of the samples. As HBV has experienced recombination events and exhibited a complicated evolution, several hypotheses have been proposed on the evolution and origins of HBV that have not yet been fully demonstrated [165]. For example, the evolutionary ratio of HBV was estimated to be approximately 10^−4^ substitutions/site/year with heterochronous samples and 10^−6^ substitutions/site/year using calibration based on HBV-human co-expansion [166,167,168]. The evolution of HBV has been estimated using ancient HBV genomes with a temporal signal [14,169]. Mühlemann et al. suggested that the root of the HBV tree was placed between 8600 and 20,900 years ago, and the substitution rate was estimated to be 8.04 × 10^−6^ to 1.51 × 10^−5^ substitutions/site/year [14]. Kocher et al. also estimated the MRCA to be ~20,000 to 12,000 years ago, and the substitution rate was estimated to be 8.7 × 10^−6^ to 1.45 × 10^−5^ substitutions/site/year using 137 ancient HBV genomes detected from Eurasian and American individuals who lived 400 to 105,000 years ago [133]. The existence of Ancient HBV reflected several known human migrations and demographic events such as the expansion of first American populations and the Neolithic transition in Europe. Several genetic events could not be expected from the human genetic and archaeological data, such as the near complete renewal of western Eurasian HBV diversity and the existence of the extinct genotype [133]. Mühlemann et al. demonstrated the evidence that genotype A, typical in modern Africa and derived from recombination, emerged outside Africa. Such a recombination event was also detected in ancient B19V genomes [73]. Ancient HBV genomes were discovered from Mexican samples derived from transatlantic slaves and might explain the introduction and dissemination of pathogens from Africa to the Americas [67,131]. In addition to the evolution of HBV within humans, reciprocal cross-species transmission might have occurred because several ancient viral lineages seem to be distinct from modern HBV lineages and show a closer relationship with non-human primate HBV strains than those of other humans [14,15].

Ancient variola virus (VARV) sequences have been discovered in several samples: a 300-year-old Siberian mummy, a 300-year-old French skeletal specimen, a 367- to 379-year-old Lithuanian child mummy, two specimens from the Czech National Museum, a 229- to 262-year-old ethanol-fixed infant leg from England, and thirteen 970- to 1400-year-old Northern European individuals [66,153,154,157,159]. Almost complete VARV genomes were obtained from samples except for in the case of the Siberian mummy and the French skeleton specimen, even though the VARV genome is relatively large, approximately 186 kb. The reconstructed genomes were subjected to phylogenetic analyses. The specimens from the Czech National Museum were first estimated to be 160-year-old samples, but they were later re-estimated as samples from the 1920s [157,158]. The MRCA was estimated to have originated roughly 371 to 508 years ago using VARV sequences derived from Lithuanian, Czech, and British samples [155,156,158,159]. Ferrari et al. estimated the evolutionary ratio as 8.5 × 10^−6^ substitutions/site/year using Lithuanian samples, and Ferrari et al. estimated it as 1.067 × 10^−5^ substitutions/site/year using Lithuanian, Czech, and British samples [155,159]. However, ancient VARV was sequenced using older samples of Northern European specimens approximately 970 to 1400 years old, indicating that the MRCA of VARV was ~1700 years ago, and the accumulation rate of nucleotide substitutions was 3.7 × 10^−6^ to 6.5 × 10^−6^ substitutions/site/year [66,160]. These estimation results differed from the ratio estimated from time-structured samples, 1 × 10^−5^ substitutions/site/year, isolated in the 20th century [170]. Since a limited number of ancient samples tends to cause incorrect estimation results, efforts to discover more ancient viral samples are necessary. In addition to the evolutionary rate estimation, Mühlemann et al. demonstrated the reduction in gene contents during VARV evolution based on the following evidence: three genes are active in modern VARV which were inactive in some or all ancient VARV, 10 inactive genes in modern and ancient VARV have different mutations suggesting parallel evolution, and the inactivation of 14 genes in modern VARV are active in some or all of the ancient VARV and eight of them encode virulence factors or immunomodulators [66]. It was suggested that the orthopoxvirus species originated from a common ancestor containing all genes present in current orthopoxviruses and that the long-term adaptation within host species caused the reduction in active genes. Babkin also revealed the differences in terms of gene contents between ancient VARV and its modern equivalent and suggested that the ancestral species contained all of the genes present in orthopoxviruses today and that long-term adaptation to within-host species occurred through a reduction of active genes [160].

Several complete sequences of ancient plant DNA and RNA viruses have been discovered despite RNA instability, as listed in Table 1 [29,34,35,36,37,38,39,40,41]. For instance, ancient barley stripe mosaic virus (BSMV) genomes were obtained from ~750-year-old barley grains [29]. The phylogenetic relationships between modern BSMV and reconstructed ancient BSMV viral genomes suggested that the divergence of BSMV most likely occurred roughly 2000 years ago. The divergence age was much older than the results based on recent serial and heterochronous sampling data. The viral lineage appears to have originated from the Near East or North Africa and spread to North America and East Asia via hosts.

Finally, we introduce a complete sequence of an ancient bacteriophage. Ancient bacteriophages were less examined in ancient specimens than viruses that infect other hosts. One study successfully reconstructed an almost complete ancient 42 kb phage genome, Siphovirus contig89 (CT89), from approximately 3800-year-old dental pulp using de novo assembly [30]. Interestingly, the reference sequence of modern CT89 was registered as only a partial sequence, around 24 kb. This suggested that the de novo assembly of ancient sequences is beneficial in the context of the reconstructing of complete ancient viral genomes, although it requires the preservation of ancient DNA. CT89 is a dsDNA phage that infects *Schalia meyeri* and is known as an oral commensal bacterium. The results of phylogenetic analysis indicated that the ancient CT89 sequences were different from the modern CT89 sequences and might reflect ancestral states.

Overall, NGS data have enabled us to obtain a vast amount of information about ancient viral genomes. When suitable viral sequences were obtained, viral genetic variations and evolution with information regarding dates could be accurately examined. In reality, limited numbers of sequences can result in inadequate estimation results, as shown in the case of VARV. Therefore, we should make more effort to increase the number of ancient samples. Recently, the number of ancient human WGS data containing viral genomes registered on the open database has increased; thus, we have a greater chance to detect ancient viral genomes from WGS data.

## 6. Long-Term Viral Evolution Reflecting Time-Dependent Rate Phenomenon (TDRP) Elucidated by Ancient Viral Sequences

Viral evolution has been inferred from the differences among current viral genomic sequencing data. Based on these differences, several evolutionary processes, including phylogenetic relationships, evolutionary rates, and divergent periods, were estimated. Recent ancient viral studies have shed light on long-term viral evolution based on age-related information regarding the nucleotide diversity between ancient and modern viruses.

Mutations cause genomic differences and genetic diversity in viruses and are maintained through several processes such as natural selection, random genetic drift, and recombination, contributing to viral evolution [171]. Therefore, analyses of viral mutations are essential for elucidating viral evolutionary history, past viral population dynamics, and viral evolutionary rates. Since several mutations are related to viral adaptation, such as when a virus acquires drug resistance and pathogenicity, it is critical to understand the evolution of viruses in terms of both its medical and epidemiological aspects in order to prevent and predict pandemics [18,172,173,174]. Several genomic and phylogenetic analyses of viral genomes suggest that viral evolution appears to be faster than that of cellular organisms because of high nucleotide substitution rates [175,176]. It is possible to observe the evolution of viral genomes using samples collected within timescales of years or by using abundant data concerning experimentally determined mutation rates [170,177]. Such results indicate that the viral evolutionary rate is determined by diverse factors including genomic architecture, replication speed, and polymerase fidelity [178]. Additionally, viral evolutionary rates vary depending on the timescale according to historical data accumulation. Both viruses and cellular organisms have shown discrepancies in the evolutionary ratio between short and long timescales, and their evolutionary rates appear to decrease with the measurement of long timescales. This phenomenon is known as the time-dependent rate phenomenon (TDRP). The TDRP has been explained by several factors such as the sequence site saturation, purifying selection, short-term changes in selection pressure, and inadequate estimation of substitution rates [179,180,181,182,183]. Furthermore, the evolutionary rates are consistent with a power-law relationship between the substitution rate and the observational period [18,180]. Ghafari et al. established a model that explains the rate decay phenomenon over a wide timescale and reproduced the ubiquitous power-law rate decay [183]. Short-term rates might reflect population-level processes such as transient deleterious mutations and short-sighted adaptations within the current host species [18]. Conversely, long-term rates reflect the actual fixation rate of mutations over historical timescales of more than thousands of years [18,184]. Therefore, ancient DNA and RNA viruses can provide valuable information about long-term viral evolution over macroevolutionary timescales. Many ancient viral studies have conducted phylogenetic analyses and estimated each virus’s evolutionary ratio and divergent events. The short-term evolutionary rates tend to be relatively fast because the divergent age might be incorrectly estimated when only modern samples are applied [181,182]. The evolutionary ratio of HBV was estimated to be roughly 10^−4^ substitutions/site/year using modern heterochronous samples, and the most recent common ancestor (MRCA) of human HBV was estimated as <~1500 years ago [165,166,168,185]. In the case of using calibration based on HBV-human co-expansion, the evolutionary ratio and MRCA was estimated at roughly 10^−6^ substitutions/site/year and 34,100 years ago, respectively [167]. As described in the previous section, ancient HBV sequences were recently discovered from human specimens that were thousands of years old [14,15]. The evolutionary ratio estimate ranged from 8.7 × 10^−6^ to 1.45 × 10^−5^ substitutions/site/year, and the root of the HBV tree was estimated to be between 8600 and 20,900 years ago [14]. Different estimation results were also obtained for VARV, as described in the previous section; the evolutionary ratio was estimated to be 1 × 10^−5^ substitutions/site/year using time-structured samples isolated in the 20th century and 3.7 × 10^−6^ to 6.5 × 10^−6^ substitutions/site/year when about 970- to 1400-year-old Eurasian samples were used [66,160,170].

Lythgoe et al. proposed that chronic viral infection with long transmission intervals requires strategies to avoid short-sighted evolution, which could be deleterious for chronic viral infections and transmission within the host population [184]. HBV has life history traits which show a slower within-host evolutionary rate compared to during replication. Lin et al. mentioned that the colonization–adaptation trade-off (CAT) model can explain the high short-term and low long-term HBV evolutionary rates [186]. When chronic HBV limited host immunity at an early stage, it favored viruses with a high replication ability. It might generate escape or adaptive mutants against the enhanced immunity infection. Then, HBV shifts to a new tolerance phase in a new host. The results mentioned that the trade-offs occurred during transmission and colonization and their effects were concentrated on nonsynonymous rather than synonymous sites. On the other hand, HIV-1 had higher within-host evolutionary rates compared to between-host [184]. The rate mismatch might be caused by temporal changes in selection pressure during infection, the frequent reversion of adaptive mutations after transmission, and the storage of viruses in the body following preferential viral transmission: in other words, “stage-specific selection”, “adapt and revert”, and “store and retrieve” [187]. Concerning the three processes, the “store and retrieve” process might be the primary factor contributing to the rate of divergence. Lythgoe et al. mentioned the importance of acute infection, and only viruses stored after acute infection are transmitted and can cause slow evolution at the population level [187]. Accumulating evidence has also suggested that minority viral populations that initiate infections have a lower evolutionary ratio within a host [184]. It might be helpful to avoid evolutionary short-sightedness through the maintenance and preferential transmission of the subpopulation of viruses that initiated the infection.

Extrapolating evolutionary rate estimation across large timescales can seriously bias analysis [182,188]. There are several arguments concerning the discrepancy and its several possible causes, such as the inappropriate use of molecular clock dating without a temporal structure, incorrect calibration points, and different replication rates [189,190]. Among them, the “lack of temporal structure” is a well-argued practical problem. Evolutionary estimation requires a detectable temporal signal demonstrated by a positive correlation among sampling times and genetic distances [191,192]. When we utilize tip dates for calibration, the population must be measurably evolving, and the sampling window must be wide enough to capture the adequate amount of genetic change [193,194]. If there is a strong disparity in the number of modern and ancient sequences, the standard test for time requires a root-to-tip linear regression for genetic distance, as a function of time might not be reliable [192,195,196]. Several methods have been established to avoid these problems caused by biased or erroneous data and evaluate the extent of the temporal structure in the datasets to be used. Firstly, the degree of the temporal signal should be calculated on time-structured datasets. It can be performed using the simple regression of the root-to-tip distance against the sampling time [169,191]. Rambaut et al. established a tool, TempEst, to visualize and analyze temporally sampled sequence data [191]. It evaluates the existence of sufficient temporal signals in the data to perform molecular clock analysis and identifies sequences with inconsistent genetic divergence and sampling dates [191]. Secondly, the date-randomization test helps to determine whether a time-structured dataset has sufficient temporal structure [169,197]. Duchêne et al. assessed a date-randomization test to investigate whether time-structured datasets had adequate temporal signals [197]. The estimated substitution rate can pass the test when the mean does not fall within the 95% credible intervals of rate estimates obtained using replicating datasets with random sampling times. One of the HBV studies used these two methods to estimate the presence of temporal signals in ancient HBV samples to increase the credibility of the estimation [14]. Tong et al. compared the different methods for estimating substitution rates using ancient DNA sequence data, regression of root-to-tip distances, least-squares dating, and Bayesian inference [193]. They recommended applying multiple methods of inference and testing for the presence of temporal signals.

## 7. Future Perspective: Detecting Highly Diverged or Extinctic Viral Genomes

Viral sequences can rapidly diverge from ancestral sequences and have little homology due to the fast evolutionary rates of viruses. Accordingly, ancient viral sequences might be difficult to detect based on homology searches because of the lack of similarities. Therefore, we need alternative methods to detect ancient viral sequences. Several viral detection methods were proposed for assembled contigs in modern viral metagenomic studies. An example is detecting viral sequences from a combination of viral gene contents and genomic structural features, including VirSorter, VirSorter2, and MARVEL [198,199,200]. Another method is using frequencies of nucleic acids or kmer-based machine-learning methods with known viral sequences, such as VirFinder [201]. The clustered regularly interspaced short palindromic repeats (CRISPR) system and prokaryotic adaptive immunological memory are also employed as nonreference-based approaches [202,203]. Bacteria can memorize the partial genomes of previously infected phages, and there are almost identical sequences between bacterial CRISPR spacers and phage protospacers [204,205]. Therefore, we can identify viral sequences utilizing bacterial CRISPR spacer sequences. Those non-homology methods might aid in detecting highly diverged or extinct ancient viral sequences. One paper indicated that viral sequences were detected from 15,000-year-old glacier ice by VirSorter [46]. Due to the low preservation quality of ancient DNA and RNA in general, there is a high hurdle for assembling contigs and searching candidate ancient viral genomes. However, if we could obtain longer contigs, the above methods would enable us to detect highly diverged or extinct viral genomes.

## 8. Conclusions

We reviewed studies on ancient viruses discovered from archaeological samples ranging from a few decades to more than thousands of years old. Thanks to advancements in sequencing technologies, several ancient viral genomes have been discovered from historical samples and utilized for evolutionary analyses. The reconstructed genomes are beneficial for obtaining epidemiological pieces of evidence from ancient times and for estimating long-term viral evolution and temporal signals. Moreover, ancient viral detection, including non-pathogenic viruses, helps to elucidate ancient viromes possibly related to ancient people’s lifestyles. Currently, the number of identified ancient viral genomes is limited; thus, efforts to detect more ancient viruses will provide more insights into viral evolution and transition from ancient times.

## Figures and Tables

**Figure 1 viruses-14-01336-f001:**
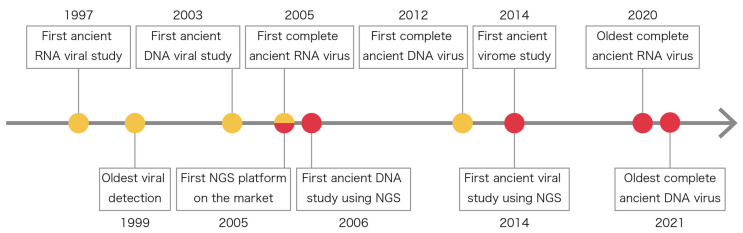
History of ancient viral studies. Yellow and pink dots indicate studies or events related to polymerase chain reaction (PCR) and next-generation sequencing (NGS) techniques, respectively.

**Figure 2 viruses-14-01336-f002:**
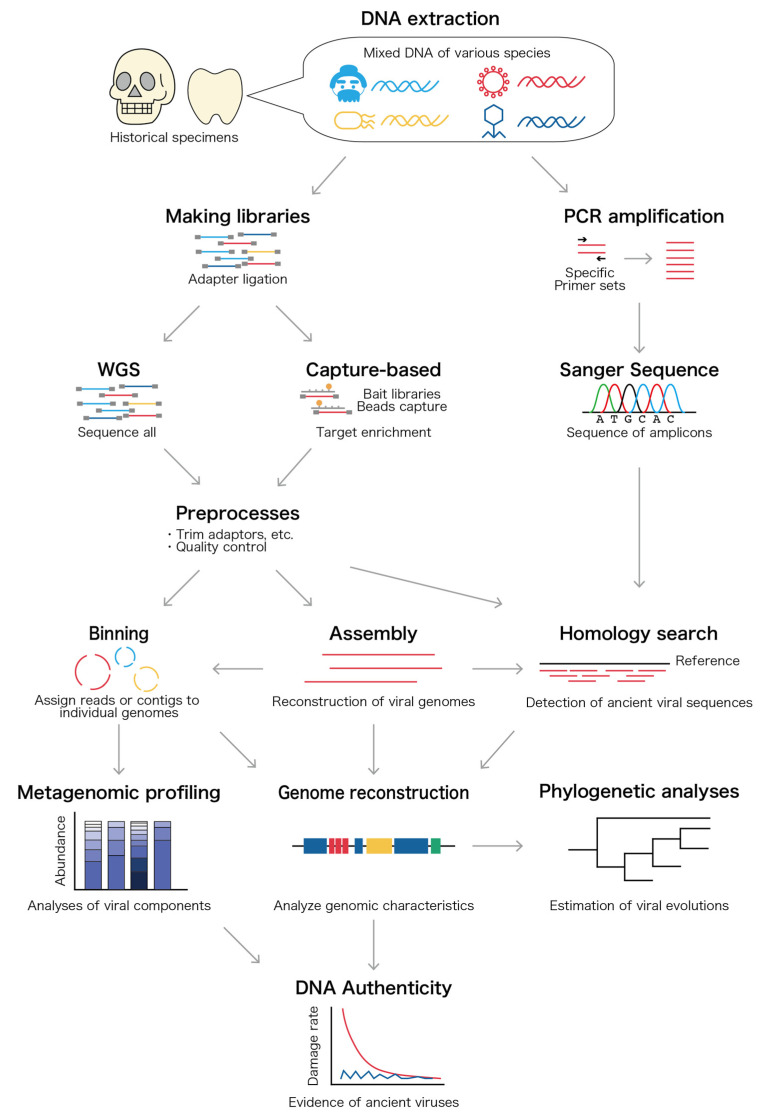
Overview of the experiments and bioinformatic analyses of ancient viral genomes. Ancient DNA can be extracted from historical specimens such as bones and teeth. The extracted DNA is derived from human, microbial, and viral genomes. Those mixed sequences can be determined by Sanger sequencing, whole genome sequencing (WGS), or capture-based sequencing based on next-generation sequencing (NGS) platforms. WGS can sequence untargeted DNA from humans, microbes, and viruses, and capture-based methods use biotinylated specific bait libraries and magnetic beads to enrich the target sequences. Following the preprocessing steps, contigs can be constructed by de novo assembly. Then, those contigs and preprocessed reads can be utilized for sequence binning to cluster the sequences into individual groups and obtain ancient viral sequences. Simultaneously, all contigs, preprocessed reads, and polymerase chain reaction (PCR) amplicons can be aligned to known viral sequences to detect candidate ancient viral sequences. Finally, the ancient viral sequences can be applied for downstream analyses: metagenomic profiling, the reconstruction of ancient viral genomes, DNA authenticity testing, and phylogenetic analyses.

**Table 1 viruses-14-01336-t001:** Overview of ancient viruses detected from historical samples.

Species Name	Type	Host	Reference Accession ID	Reference Length (kb)	Detected Length (kb) ^1^	Method	Sample	Sample Age (ya) ^2^	Region	Accession ID of Ancient Viral Genomes	Accession ID of Raw Reads	References
African cassava mosaic virus (ACMV)	ssDNA ^3^	*Manihot glaziovii*	NC_001467, NC_001468	5.5	5.5	PCR ^10^, WGS ^11^	Leaf of Manihot glaziovii specimen	94	Bambari, Central African Republic	MW788219, MW788220	PRJNA698751	Rieux et al., 2021 [41]
Anelloviridae	ssDNA	*Homo sapiens*	AB303563	3.2	0.078	PCR	Dental pulp	200	Kaliningrad, Russia	NA	NA	Bédarida et al., 2011 [128]
Ancient caribou feces-associated virus (aCFV)	DNA	Plant	NA	NA	2.2	PCR	Coprolite	700	Northwest Territories, Canada	KJ938716	NA	Ng et al., 2014 [43], Holmes, 2014 [129]
Ancient Northwest Territories cripavirus (aNCV)	+ssRNA ^4^	Insect	NA	NA	1.8	PCR	Coprolite	700	Northwest Territories, Canada	KJ938718	NA	Ng et al., 2014 [43], Holmes, 2014 [129]
Barely stripe mosaic virus (BSMV)	+ssRNA	*Horedum vulgare*	NC_003469, NC_003481, NC_003478	10.2	10.2	WGS	Barely grain	750	Upper Nubia, Egypt	NA	NA	Smith et al., 2014 [29]
Barely yellow dwarf virus (BYDV)	+ssRNA	*Avena fatua, Danthonia californica, Glyceria elata, Koeleria macrantha, Phalaris coerulescens*	NC_004750	5.7	3.6	RT-PCR ^12^	Herbarium specimens	82–105	US	DQ115532–DQ115534, DQ118372, DQ631844–DQ631846, DQ631856, DQ631857	NA	Malmstrom et al., 2007 [39]
Citrus leprosis virus (CiLV)	+ssRNA	*Citrus aurantium, Citrus sinensis*	NC_008169, NC_008170	13.7	12.7	WGS	Herbarium specimens	55–90	US, Mexico, Argentina, Brazil	KT187687–KT187693	NA	Hartung et al., 2015 [40]
Epstein–Barr virus (EBV)	dsDNA	*Homo sapiens*	NC_007605	171.8	22.9	WGS	Chewed birch pitch	5700	Island of Lolland, Denmark	NA	PRJEB30280	Jensen et al., 2019 [47]
Hepatitis B virus (HBV)	dsDNA-RT ^5^	*Homo sapiens*	NC_001611	3.2	3.2	PCR	Liver of a mummy	500	Yangju, Korea	JN315779	NA	Kahila Bar-Gal et al., 2012 [33]
					3.2	Capture ^13^	Distal femur, skin, muscle of a mummy	500	Naples, Italy	MG585269	NA	Patterson et al., 2018 [130]
					3.2	WGS, Capture	Tooth cementum, petrous bones	822–4488	Central to western Eurasia	ERS2295383–ERS2295394	PRJEB9021, PRJEB20658	Mühlemann et al., 2018 [14]
					3.2	WGS	Teeth	340–5000	Germany	NA	PRJEB24921	Krause-Kyora et al., 2018 [15]
					2.9	WGS, Capture	Tooth	396–569	Mexico City, Mexico	NA	PRJEB37490	Barquera, et al., 2020 [131]
					3.1	WGS	Soft tissue and bone of a mummy	2000	Abusir el-Meleq, Egypt	NA	PRJEB33848	Neukamm et al., 2020 [132]
					3.2	WGS, Capture	Tooth root	500	Mexico City, Mexico	MT108214	Available at Dryad ^14^	Guzmán-Solís et al., 2021 [67]
					3.2	WGS, Capture	Teeth, bones, petrous bones	400–10,500	Eurasia and US	ERS6597748–ERS6597884	PRJEB45699	Kocher et al., 2021 [133]
Hepatitis C virus (HCV)	+ssRNA	*Homo sapiens*	NC_004102	9.7	0.336	RT-PCR	Archived blood samples	69	US	KF261594, KF261595	NA	Gray et al., 2013 [134]
Human immunodeficiency virus type 1 (HIV-1)	ssRNA-RT ^6^	*Homo sapiens*	NC_001802	9.2	~0.3	RT-PCR	Plasma samples	63	Kinshasa, Democratic Republic of Congo	NA	NA	Zhu et al., 1998 [26]
					8.6	RT-PCR	Frozen serum samples	50	New York City, US	KJ704787–KJ704797	NA	Worobey et al., 2016 [135]
					8.3	RT-PCR, amplicon sequence	Formalin-fixed paraffin-embedded tissues	56	Kinshasa, Democratic Republic of Congo	MN082768	NA	Gryseels et al., 2020 [136]
Human papillomavirus (HPV)	dsDNA ^7^	*Homo sapiens*	NC_027779	7.3	0.141	PCR	Mummy of a Renaissance noble woman	454	Naples, Italy	NA	NA	Fornaciar et al., 2003 [137]
Human parvovirus B19 (B19V)	ssDNA	*Homo sapiens*	NC_000883	5.6	0.275	PCR	Long bones	92	Karelia district, Finland	NA	NA	Toppinen et al., 2015 [138]
					5.9	WGS	Dental, skeletal remains	500–6900	Eurasia, Southeast Asia, Greenland	NA	PRJEB26712 ^15^	Mühlemann et al., 2018 [73]
					4.4	WGS, Capture	Tooth roots	500	Mexico City, Mexico	MT108215–MT108217	Available at Dryad ^14^	Guzmán-Solís et al., 2021 [67]
Human T-cell leukemia virus type 1 (HTLV-1)	ssRNA-RT	*Homo sapiens*	NC_001436	8.5	0.316	PCR	Mummy	500	Andean, US	NA	NA	Li et al., 1999 [139], Gessain et al., 2000 [140], Vandamme et al., 2000 [141]
Influenza A virus	-ssRNA ^8^	*Homo sapiens*	NC_026431-NC_026438	13.2	12.7	RT-PCR	Formalin-fixed paraffin-embedded lung tissues	104	US	AF116575, AF250356, AF333238, AY130766, AY744935, DQ208309–DQ208311	NA	Taubenberger et al., 1997 [12], Reid et al., 1999 [142], Reid et al., 2000 [143], Basler et al., 2001 [144], Reid et al., 2002 [145], Reid et al., 2004 [146], Taubenberger et al., 2005 [147]
					12.7	RT-PCR, WGS	Formalin-fixed paraffin-embedded lung tissues	104	New York City, US	NA	PRJNA178740	Xiao et al., 2013 [148]
Measles morbillivirus (MeV)	-ssRNA	*Homo sapiens*	NC_001498	15.8	15.8	WGS	Formalin-fixed paraffin-embedded lung tissues	110	Berlin, Germany	NA	PRJEB36265	Düx et al., 2020 [149]
*Mollivirus sibericum*	dsDNA	*Acanthamoeba castellanii*	NA	NA	651	WGS	Permafrost layer	30,000	Northeast Siberia, Russia	KR921745	NA	Legendre et al., 2015 [45]
Papillomavirus	dsDNA	*Neuroma cinera*	MF416381	7.4	0.677	PCR	Unwashed midden materials	27,000	Arizona, US	MH136586, MH136587	NA	Larsen et al., 2018 [150]
*Pithovirus sibericum*	dsDNA	*Acanthamoeba castellanii*	NA	NA	610	WGS	Permafrost layer	30,000	Northeast Siberia, Russia	KF740664	NA	Legendre et al., 2014 [44]
Potato virus X (PVX)	+ssRNA	*Solanum tuberosum*	NC_011620	6.4	0.75	RT-PCR	Freeze dried leaves	38–52	Australia, England	GU384732–GU384734, GU384737–GU384738	NA	Cox and Jones et al., 2010 [35]
Potato virus Y (PVY)	+ssRNA	*Solanum tuberosum*	NC_001616	9.7	9.7	RT-PCR	Potato	84	Netherlands	EU563512	NA	Dullemans et al., 2011 [36]
					9.7	WGS	Freeze-died PVY cultures	38–79	UK	KP691317–KP691330, MT200665–MT200668	NA	Kehoe and Jones, 2016 [37], Green et al., 2020 [38]
Simian T-lymphotropic virus type 1 (STLV-1)	ssRNA-RNA	*Cercopithecus aethiops*	MF622054	8.4	0.467	PCR	Skeletons	122	Central Africa	NA	NA	Calvignac et al., 2008 [151]
Siphovirus contig89 (CT89)	dsDNA	*Schalia meyeri*	KF594184	2.4	4.2	WGS	Dental pulp	3800	Hokkaido, Japan	LC585292	PRJDB7235	Nishimura et al., 2021 [30]
Tomato mosaic tobamovirus (ToMV)	+ssRNA	Dicotyledonous, monocotyledonous	NC_002692	6.4	0.347	RT-PCR	Ice cores	<500–140,000	Greenland	NA	NA	Castello et al., 1999 [152]
Variola virus (VARV)	dsDNA	*Homo sapiens*	NC_001611	185.6	0.718	PCR	Pulmonary tissue of a mummy	300	Siberia, Russia	JX080525–JX080527	NA	Biagini et al., 2012 [153]
					0.43		Skeleton	300	Marseille city, France	NA	NA	Meffray et al., 2021 [154]
					166.8	Capture	Soft tissue of a mummy	367–379	Vilnius, Lithuania	KY358055, BK010317	PRJNA348754	Duggan et al., 2016 [155], Smithson et al., 2017 [156]
					185.4	WGS	Forefoot and piece of skin	100	Prague, Czech	LT706528, LT706529	PRJEB18730	Pajer et al., 2017 [157], Porter et al., 2017 [158]
					158.1		Ethanol-fixed infant leg	229–262	London, England	NA	PRJEB35140	Ferrari et al., 2020 [159]
					192.3	WGS, Capture	Skeletons	970–1400	Northern Europe	LR800244–LR800247	PRJEB38129	Mühlemann et al., 2020 [66], Babkin et al., 2022 [160]
Vaccinia virus (VACV)	dsDNA	*Homo sapiens*	M35027	191.7	184.7	WGS, Capture	Vaccination kits	156	Philadelphia, US	MN369532	PRJNA561155	Duggan et al., 2020 [48], Brinkmann, et al., 2020 [49], Duggan et al., 2020 [50]
Zea may chrysovirus 1 (ZMCV1)	dsRNA ^9^	*Zea mays*	NA	NA	11.3	WGS, RT-PCR	Maize cobs	1000	Antelope house, US	MH931189–MH931208, MH936006, MH936007, MH936014–MH936017	NA	Peyambari et al., 2019 [34]

^1^ the longest length within reconstructed viral sequences, ^2^ years ago (ya), ^3^ single-strand DNA virus (ssDNA), ^4^ positive-strand RNA virus (+ssRNA), ^5^ double-strand DNA virus (dsDNA), ^6^ retro-transcribing DNA virus (dsDNA-RT), ^7^ retro-transcribing RNA virus (ssRNA-RT), ^8^ negative-strand RNA virus (-ssRNA), ^9^ double-strand RNA virus (dsRNA), ^10^ polymerase chain reaction (PCR), ^11^ whole genome sequencing (WGS), ^12^ reverse transcription polymerase chain reaction (RT-PCR), ^13^ capture-based sequencing (capture), ^14^ NGS reads are available at Dryad Digital Repository: https://dx.doi.org/10.5061/dryad.5x69p8d2s accessed on 18 June 2022, ^15^ No public data are linked to this project (2 June 2022). The alignments and XML files used to perform the analysis presented in this paper are available at https://github.com/acorg/parvo-2018 accessed on 18 June 2022. NA stands for not applicable.

## Data Availability

The data presented in this manuscript are available in Table 1.

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
