# Peer review of "Detection of Ancient Viruses and Long-Term Viral Evolution"

_viruses, 2022, doi:10.3390/v14061336_

Round 1

Reviewer 1 Report

This article reviews the detection of ancient viral sequences and their use to understand the long-term viral evolution. Generally, the article is interesting and provides important resources. However, some points described below require attention.

The English language needs some revision, particularly in the abstract. There is somewhat repetition and overlap in the text that can be avoided (see, for example, "Additionally" twice declared).

Section 3.2 In addition to the mentioned limitations of PCR, DNA fragmentation should also be included as it dramatically affects detection success. This characterization is mentioned in section 5, disconnected from the other paragraphs, could be brought up to section 3.2.

Also, in section 3.2, the Authors mention WGS as a good approach for unbiased pathogen detection, which is correct. However, they do not mention that the sensitivity of this approach is low and fails to deliver, in most cases, meaningful sequencing data for pathogen characterization and study of evolution. Indeed most works use WGS for screening, but this is followed by capture.

The authors could mention the challenges of verifying the authenticity of early works on ancient viruses by PCR. For example, the finding of HTLV in an Andean mummy by Li et al. has been questioned as contamination by Vandamme et al. Nat Med vol6 num 3 march 2000. The authors could elaborate on the discrepancy.

The damage rate of viruses may contain different patterns depending on the virus, for example, from the mitochondrial DNA patterns. Please, consider commenting on this.

Perhaps it would be helpful to add comments on the USER treatment and degradation-deamination patterns. See, for example, "Ancient DNA analysis" from Orlando et al. (also referred by this article).

The authors could elaborate more on the messages drawn by the different studies other than just MRCA and substitution rates. For example, regarding VARV, Mühlemann et al. report on 14 Immunomodulatory and host range genes, which were subsequently lost in modern VARV. They suggest that the smallpox virus evolved from more minor pathogenic forms during the Viking era to the highly virulent strains of modern times. Whether these ancient viruses represent true primeval lineages, instead of being transitional (e.g., representing different jumps from animals to humans), pends on finding similar or even older strains.

It is sometimes hard to follow whether the interest of the review is on the human virome, as mentioned in the introduction, or on ancient viruses in general. There is quite a mix-up in the text. For example, mentioned are caribou feces associated virus isolated from caribou feces or Barely Stripe Mosaic Virus is not part of the human virome.

In Figure 1, the first application of NGS to ancient remains in 2006 could be present as an additional NGS event.

In Figure 2, removing human reads when analyzing ancient DNA or RNA viruses may be problematic (depending on the virus) because several viruses contain similar sequences, such as Herpesviruses to nuclear human DNA. Nevertheless, maintaining the host reads comes at a higher computational cost. The authors could comment on this.

Because (usually) there is high degradation and fragmentation of ancient DNA sequences, the de-novo reconstruction with efficiency is rare. Therefore, reference sequences play a crucial role in the metagenomic identification of viral genomes from ancient samples. For screening large amounts of data from WGS, the mappers or aligners must be very sensitive and fast. Both are not compatible because it is a trade-off. One must identify the viruses potentially contained in the samples because blasting or aligning all the viral references on time is prohibitive. Therefore, while searching for metagenomic viral content, the identification of viral sequences in the samples must be primarily applied by the metagenomic binning tools (for example, with kraken2). According to this and section 3.3, the order of first describing the aligners and then the metagenomic mappers may be confusing. Usually, the aligners are essential to build a consensus sequence and apply the variant call only after the binning or metagenomic inference. However, sometimes they are also used after for classification purposes.

Please check if it adds value to the work to mention TRACESPipe. TRACESPipe is the first pipeline to reconstruct viral genomes by combining multiple organs (for example, hard tissues such as bone and teeth in the case of aDNA); it also uses de-novo and reference-based assembly approaches.

Setting a complete genome at the same level as one contig or a set of contigs is not fair without providing detailed information. It would be interesting to have information on the present size of similar genomes and the reconstructed ancient ones in the table. Although there is recombination and other events, it may be helpful to have a notion of the degree of reconstruction completeness.

Please consider adding information to the community of the IDs or sequencing runs information on how to access the project databases with these FASTQ reads (from table 1). Also, if the reads are not publicly available, provide that information. This feature would be beneficial to readers and add substantial value to the article, for example, to computational paleovirologists or people who are now initiating these studies/research.

Other problems with the difference between molecular clocks are potential ambiguities in the reconstruction process, such as repetitive parts - for example, as an extended region of repetitive DNA in the EBV or the hairpins in the B19V. Also, different depth coverage between reconstructions, untrimmed tips or low-quality tip reads, and different reconstruction methodologies from different groups may provide differences in these clocks. It would be interesting to have comments on these issues.

Please, change the word order from "Most common recent ancestor (MRCA)" to "Most Recent Common Ancestor";

Reviewer 2 Report

The ancient viruses derived from historical remains provide useful information for our understanding of long-term virus evolution, past epidemic, and life style of ancient individuals. Nishimura et al conduct a comprehensive review on detection of ancient virus and viral evolution. The paper is well written. I only have one comment.

As mentioned by the authors, several factors, including mutation saturation, purifying selection, and change in selection pressure, may cause time-dependent rate phenomenon (TDRP). For chronic infected virus, such as HBV, the selection regime within and between hosts may be different. On the one hand, to continually spread among hosts, the virus must maintain high transmissibility and replicative ability. On the other hand, the virus must adapt to the local environment within a host during the course of infection, even at the cost of lowered transmissibility. Selection pressures during transmission/colonization and the course of infection can differ markedly, particularly for chronically infecting viruses. I suggest the authors to further explore this issue in the cause of TDRP.  

Reviewer 3 Report

Manuscript review: viruses-1729376

Nishimura et al., Detection of ancient viruses and long-term viral evolution

General comments: This manuscript summarizes studies to isolate viral DNA/RNA sequences from ancient sources, the methodology used for these studies, and conclusions derived from these studies. The review represents a good beginning, but has a number of characteristics that reduce enthusiasm for the material. The review seems superficial, and would benefit from additional attention to detail. In some instances, the authors fail to add important experimental details that would be very useful for the reader. The authors make generalized comments that would benefit from specific examples to illuminate these comments, but these are often lacking. See below for some suggestions, which illustrate the problem, but are not an exhaustive list of these problems. In addition, the text of the manuscript seems stilted and is often worded in awkward ways that detract from the message that the authors seem to want to convey. It is strongly suggested that a good copy editor should go over this manuscript to clean up the English language usage.

Specific comments:

1. pg. 3, section 3.1: The second paragraph of this section is almost uninterpretable due to the English language used. I am not sure what the authors are trying to say here.

2. pg. 4, first paragraph, top of page: This paragraph discussed the necessity to avoid contaminants in ancient DNA preps. Missing from this discussion is any mention of how the experimenter should go about assessing contamination in these ancient DNA preps, and how to modify protocols to prevent these from being introduced.

3. pg. 4, section 3.3: The last sentence in this paragraph makes no sense. This is a brief discussion of bioinformatics tools to be used for viral DNA analysis. The authors state "Arimendi Cardenas et al. reported that centrifuge would be the most suitable tool for human DNA virus...". A centrifuge is not a bioinformatics tool. Please clarify this.

4. pg. 6, paragraph at the top of page: This section discusses the necessity to assess damage to ancient viral DNA; more description of these alterations in the DNA, and consequences for sequencing, would be helpful to the reader. The authors propose software to aid in this evaluation. Some mention of how this software works also would be useful for the reader.

5. pg. 6, first paragraph in section 4. The authors state that "some differences existed between the coprolite virome and modern human stool virome at the taxonomic level, but were functionally more similar". A description of these differences would benefit the reader, along with a better explanation of why these differences appeared not to be important. 

6. pg. 6, second paragraph in section 4. The last sentence in this paragraph states "Those viral components might be related to several factors, including dietary habits, culture, and health status". This sentence is typical of many that the authors make in this review, speculating that ancient virome analysis could be used for discerning important facts about the individuals harboring these viromes. However, I do not fine anywhere in the text where this speculation is drawn out by examples from the literature. This would be useful information for the reader.

7. pg. 8, last sentence of second paragraph: The authors state "Therefore, we should pay attention when estimating phylogenetic relationships and evolutions based on a few fragmented sequences". So, are there examples where inappropriate analyses have led to incorrect conclusions? How were these inconsistencies corrected? What 'rules of thumb' should investigators use to avoid such problems in the future?

8. minor comment: In several instances, the authors do not define acronyms that are used in the text. For instance, on page 9, the authors introduce 'HBV' and 'VARV' without defining these. What viruses to these acronyms refer to?

Round 2

Reviewer 1 Report

The authors have addressed my suggestions. Thank you!
In my opinion, the article is of very high quality.
Below I include some minor suggestions.

In Figure 1, the dates of, for example, "Oldest viral detection - 1999" are cut. I'm unsure if it is only my visualizer or if the authors can also see it. If you do, please, ignore this.

If it adds value, please check out [https://doi.org/10.3390/genes9090445]. This article describes a methodology for metagenomic composition identification and is applied in a sample of an ancient polar bear from Svalbard with 130k years [https://doi.org/10.1111/j.1751-8369.2008.00087.x]. Along with much other content (for example, a phage), an endogenous retrovirus has been found in the samples. Since there was human contamination on the samples, the question of the endogenous retrovirus is puzzling, namely if the origin is from a human, the bear, or both reads (there is some similarity between both retroviruses, and some of the reads are very small). For example, this work [ https://doi.org/10.3390/v7112927] reports the presence of the polar bear retrovirus and associated it with the evolution of the virus.

Reviewer 3 Report

In response to the previous review, this manuscript has been extensively edited for English language, and significant new information has been added to address previous concerns about the somewhat superficial nature of the earlier manuscript. These modifications have improved the manuscript considerably.

Author Response

Thank you for stimulating us to strengthen our manuscript with your valuable comments and queries. We have worked hard to incorporate your inputs.